# Design of an Impedance-Controlled Hot Snare Polypectomy Device

**DOI:** 10.3390/s20010142

**Published:** 2019-12-24

**Authors:** CurtisLee Thornton, JungHun Choi

**Affiliations:** Department of Mechanical Engineering, Georgia Southern University, Statesboro, GA 30460, USA; ct05456@georgiasouthern.edu

**Keywords:** electrosurgical unit, polypectomy, bioelectrical impedance, snare

## Abstract

This paper goes through the process of first designing a feedback system that allows for the measuring of impedance while using the hot snare polypectomy method. The electrosurgical unit used in this study was the Olympus PSD-30. After the impedance-controlled feedback system was completed, the device was tested under a range of power settings from 10 W–50 W. The test was performed ex vivo using porcine colon samples. Using the information gathered from these tests, a technique of determining the threshold of perforation and implementing a system to automatically stop the applied current from the PSD-30 was developed. The data showed that after an increase in impedance of 25% from that of the initially measured impedance, perforation ensued in the tissue samples. Using this information, the device was programmed to interrupt the PSD-30 at this threshold point. This final design was tested and proved able to automatically prevent the event of perforation from occurring, resulting in the ability to prevent serious complications.

## 1. Introduction

In 2015 alone, there were 17.5 million cancer cases, including 8.7 million deaths worldwide [1]. This places cancer as the second leading cause of death worldwide. This trend also carries over to the United States. In the same year, there were over 2.7 million total deaths, with cancer ranked second, contributing to 22% of the total [2]. Of all the types of cancers, colorectal cancer was the fourth most commonly diagnosed cancer, contributing to over 50,000 deaths each year [3]. The risk of developing colorectal cancer is 1 in 22 for men and 1 in 24 for women [4]. Despite being the fourth-highest killer, colorectal cancer has seen a steady decline in the incidence and mortality rates since the 1980s. This is in part contributed to the improvements in treatment as well as the increase in the number of screenings [5]. The main avenue of approach for screening consists of a colonoscopy. When a polyp is found, polypectomy needs to take place. It has been found that the use of polypectomy during the colonoscopy has significantly reduced the incidence of and mortality from colorectal cancer [6], although multiple complications can arise from this procedure. Some of these include post-polypectomy coagulation syndrome, bleeding, and the most severe, perforation. Although the rates for colorectal cancer have been decreasing for decades, the perforation and mortality rate has remained stable from 2001 to 2015 [7]. This is alarming, as perforation can cause severe post-surgery complications and even death. One study, of over 250,000 performed colonoscopies, showed a perforation rate of 0.07% [8]. Another study of nearly 38,000 colonoscopies showed a perforation rate as high as 0.113%, with a mortality rate of 25.6% of those that underwent surgical intervention [9]. It was concluded by Panteris et al. that “awareness and experience are the only preventative measures that can limit the incidence of perforation” [10]. If the experience of the surgeon is a major factor in the rate of post-polypectomy complications, then there needs to be a way to assist the surgeon further to continually improve the treatment for the patient. To accomplish this, we need to first investigate the methods used and the factors that contribute to the tissue damage. From this information, we can then design a device that will lower the need for an experienced surgeon. This will, in turn, reduce the number of complications and continue to lower the overall perforation and mortality rate.

The polyp sizes most commonly encountered (over 80–90%) during a routine colonoscopy include diminutive (≤ 5 mm) and small (6–9 mm) [11]. The main methods used to remove these polyps include the use of either forceps or a snare. A survey of 189 gastroenterologists concluded that the use of forceps was preferred for diminutive polyps while the use of an electrosurgical snare was the preferred method for removing small polyps [12]. The shift towards snare polypectomy for even diminutive polyps was reported, as the use of forceps is associated with incomplete removals of the polyp as well as higher complication rates [6]. Others trying to standardize the polypectomy technique also agree with using the snare for polyps < 10 mm [13]. It was also shown by Singh et al. that during hot snare polypectomy, the electrosurgical current used included coagulation 46% of the time, a blend 46%, pure-cut 3%, and 4% varied the current [12]. For these reasons, this study will focus on the use of the hot snare polypectomy technique using the coagulation mode for the current.

Before a device can be designed, the factors that contribute to the tissue damage caused by electrosurgery need to be explored. Many details of the fundamentals of electrosurgery are discussed by Morris et al. [14]. A high-frequency alternating current, supplied by an electrosurgical unit (ESU), is passed through the tissue. This thermal energy heats the water inside of the cells and ruptures the cell membrane. The main factor that contributes to tissue damage is the current density, which is the measure of the intensity and concentration of the current. The variables that affect the current density include some of which can be controlled by the user and some which are based on the patient. The variables that can be controlled by the user include the length of time the current is applied to the site, the waveform, and the power applied. During this study, the waveform was set to coagulation mode, as discussed before, and the power and length of time applied to the site were varied. The variables which are out of control of the user and that which vary from patient to patient include the total impedance, from the size of the polyp to the size of the patient. Impedance is the effective resistance of an electric circuit. There is a relationship in the change of impedance in the tissue and the damage done at the ablation site [14]. As the thermal damage increases at the ablation site, so too does the impedance. The ablation site is where perforations and bleeding can cause severe post-surgery complications and even death. The idea of monitoring the change in impedance during the operation of the ESU is not a new idea. Multiple big manufactures of ESU’s have this integrated into their devices. It is more commonly seen however for the bipolar method, which both sending and receiving electrodes of the current flow through the accessory [14]. The difference in this study is the fact that this is done using a hot snare, which uses the monopolar method. This brings on additional complications in the accuracy of the measurements due to the length of the wire of the snare and grounding patch easily being over two meters and having to integrate the patients’ body, not just the polyp, into the measurements. In this case, the circuit would be the path the current takes from the ESU through the snare, then through the polyp and patient, and back to the ESU through the ground patch. If we are able to understand and measure the impedance to determine at which point there is too much tissue damage and control the ESU to stop before serious damage occurs, then we can reduce the experience need of the surgeon and lower the number of complications arising during electrosurgery. In this paper we design, build, and test a device that does just that.

## 2. Materials and Methods

The development of this device requires two main steps to be performed before the safe and proper implementation of the final product. These steps include the development of the impedance-controlled feedback system and validation of the safe and proper operation of the system.

### 2.1. Development of Feedback System

To accomplish the task of developing a device that can measure and track the changes of the impedance over time while the ESU is in operation requires three separate systems that need to be created and integrated together. The three parts of this feedback system include the modified ESU, a measuring system, and a controlling system. The ESU modified in this study, which supplies the high-frequency alternating current, is the Olympus PSD-30 (Olympus, Shinjuku, Tokyo, Japan). The PSD-30 has a fundamental frequency of 350 kHz with an open-circuit output voltage upwards of 900 V. The ESU is capable of outputting power ranging from 2 W–50 W. The ESU also has three different methods of operation for each the coagulation and cutting mode. The three modes for coagulation include soft, auto-stop, and forced. The soft mode is used to stop light bleeding as well as marking tissue where very little tissue carbonization occurs in this mode. The auto-stop mode uses soft coagulation and is used to reduce tissue carbonization and invasion while using forceps. The forced mode is used for strong coagulation and hemostatic effects. The three modes used for cutting include pure, blend1, and blend2. The pure mode uses the cut wave-form with no hemostatic effects while the blend1 mode contains some hemostatic effects and blend2 contains even more hemostatic effects than blend1. The measuring system is comprised of a computer running LabVIEW (National Instruments, Austin, TX, USA) that is used to calculate the impedance values and determine at what point the ESU should be interrupted. The final piece of the puzzle consists of the controlling system. This control system uses an Arduino UNO (Arduino LLC, Ivrea, Italy) microcontroller to coordinate the safe measurement of the impedance. With each of these three systems synchronized, accomplishing the task of initiating, measuring, and interrupting the ESU based on the change in impedance value can be developed.

The modification of the ESU was accomplished by way of the foot pedal assembly. This allows for the reading of when the user presses down on the assembly to initiate the firing, as well as a place to stop the ESU from firing by interrupting the signal sent to it. The foot pedal assembly works by use of two normally open momentary switches as shown in Figure 1a. One switch is used to control the cutting mode, while the other switch is used to control the coagulation mode. The modification was accomplished by way of these two switches. The wires connecting from the switches were cut, and then the microcontroller was placed in between each wire as shown in Figure 1b. This gave the microcontroller two inputs and two outputs. When one of the pedals are pressed, a 5 V signal is passed, on the other hand, when the pedals are depressed, the signal reads 0 V. This allows for the microcontroller to determine the state of the switch as either digital high, the user is pressing the button to fire, or digital low, the user is not pressing the button. With this information, the microcontroller can tell the ESU to fire or not. Essentially the user presses the button on the foot pedal assembly telling the microcontroller they wish to fire the system, and then the microcontroller tells the ESU whether to fire or not, effectively playing the middleman.

The second portion of the feedback system comprises the measuring system (Figure 2). This part of the feedback system is used to take the impedance measurement of the device under test (DUT), which in this case is the model the alternating current from the ESU passes through. The measuring system consists of multiple parts which each play a vital role in the operation. The first of which is the E3633A benchtop DC power supply (Keysight Technologies, Santa Rosa, CA, USA) which is used to power the PYB10-Q24-D5 DC-DC converter (CUI Inc, Tualatin, OR, USA). The DC converter takes an input voltage range from 9V–36V and was chosen because it can support an output current from ±50 mA to ±1000 mA and supplies the ±5 V required by the Howland current pump, buffers, and relays. Note that the power connections for each of these components are not shown in the diagram to keep the figure from being cluttered. The Howland current pump was chosen for its’ ability to provide a stable output current over varying input frequencies which is required for accurately measuring both the DUT and reference resistor. The current pump uses the Voltage Controlled Current Source (VCCS) method and was tested and confirmed for accuracy over a range of 3 kHz–1 MHz to provide 200 µA RMS by measuring the voltage from the reference resistors. The current has a constant value regardless of the DUT using OPA655 operational amplifiers. These op-amps were chosen as they outperformed three others in a study by Bertemes-Filho et al. [15].

In addition, a B&K Precision model 4053 Function Generator (B&K Precision Corp, Yorba Linda, CA, USA) is used to generate the 50 kHz 1 V_pp_ sinusoidal wave that is used as the reference signal. The 50 Ω output impedance and varying current from the function generator are handled by the current pump to ensure a constant current for measuring. The single 50 kHz frequency was chosen for two reasons. The first of which has to deal with timing. Although multifrequency bioimpedance spectroscopy is the preferred method, as it allows for a complete picture of the properties of the subject [16], this method takes too long to be practical in this application. To perform a reasonable multifrequency measurement, frequencies from about 5 kHz to 500 kHz need to be measured, which dismisses this technique. This frequency range is the most commonly used for bioimpedance analysis [16]. Knowing that a single frequency measurement will be required, leads to the second reason. The 50 kHz frequency is the most commonly used frequency when performing a single frequency bioelectrical impedance measurement. This is due to the weighted sum of extra-cellular water (ECW) and intracellular water (ICW) resistivities at this frequency [16]. With knowing that the current source supplies a constant current of 200 µA_pp_ RMS at this reference frequency, the impedance of the DUT and reference resistor can be calculated. To calculate these impedance values, two voltage output signals are collected with a National Instruments NI 6353 data acquisition (DAQ) device as shown in Figure 2, where the reference resistance is 475 Ω. This resistance value was chosen as it is a comparable resistance to the DUT. This allows for a measurable voltage drop that is not too small where the error of the measuring device does not become a problem. Before the signals make it to the DAQ however, they are passed through a buffer, also known as an active electrode (AE) Figure 3, and then through an instrumentation amplifier (IA) with a gain of two. The buffer is used to stabilize the signal, while the amplifier is used to increase the small output voltage being received by the DAQ for higher resolution. These voltages collected by the DAQ are then read into LabVIEW via the computer. The sampling frequency of the DAQ is set to 251 kHz. This adheres to the Nyquist rate and ensures the samples are not a constant multiple of the measuring frequency ensuring that the full waveform is measured. The 475 Ω reference resistor is used to confirm that the voltage measurements taken of the DUT are accurate. To ensure this, the measured voltage drop should be an ideal 268.7 mV, which relates to a resistance of 475 Ω. This value is derived from multiplying the current from the pump of 282.84 µA with the known reference resistance of 475 Ω, and then multiplying by a factor of two for the amplifier. With this measurement, the system can be said to accurately measure the voltage drop across the DUT.

Now that the modification of the ESU and measuring system is completed, the control system part of the feedback system needs to be developed (Figure 4). For the control system, a microcontroller is used to coordinate the firing, measuring, and the on and off synchronization of the relays. The microcontroller utilizes three inputs and five outputs to achieve this. Two inputs are used for reading the state of the foot pedal assembly, one for coagulation and one for cut mode, while the third input is connected to the DAQ which is attached to an interrupt pin on the microcontroller. This third input is used by LabVIEW to stop the ESU before perforation occurs based on the measured impedance and calculated threshold value. The five outputs include one for each of the two sets of two CPC1988 solid-state power relays (IXYS, Milpitas, CA, USA). These high-power solid-state relays are used to separate the DUT between the measuring system and the modified ESU. They were chosen because of the high 1000 V_p_ blocking voltage [17] which is required to block the maximum output voltage of about 900 V [18] that the PSD-30 ESU is capable of. Two more outputs are used to control each of the 9002 SIP reed relays (COTO Technology, North Kingstown, RI, USA) which are used to send a 5 V signal to the modified ESU foot pedal assembly to fire either the coagulation or cut mode. As the signal required to fire the ESU is low voltage, the only concern was that of speed. That is why the 9002 SIP reed relay was chosen, it has an operating time of 0.35 ms and a release time of 0.1 ms [19]. It is important to note that each of these four outputs is controlling the relays through NPN bipolar transistors configured to perform as a switch. This was done to allow the relays to be powered by the DC-DC converter’s 1000 mA power supply, as the microcontroller is not able to source enough current for proper switching. The transistors are not included in the diagram for simplicity. The final output is connected to the DAQ and tells LabVIEW whether it is acceptable to measure or not. This output is especially important as the feedback system should not take measurements while the ESU is firing as the measuring relays are open and an infinite impedance measurement would be calculated.

Now that the feedback system is complete, the operation of the system is as follows. The complete impedance feedback system is shown in Figure 5.

As the user presses the foot pedal, a 5 V signal is sent to the microcontroller indicating either the coagulation or cut switch has been pressed. The microcontroller reads this input and starts by opening the solid-state power relays that connect the measuring system to the DUT, which will be referred to as the ‘measure relays’. While this is happening, the solid-state power relays connecting the ESU to the DUT are closed, these relays will be referred to as the ‘ESU relays’. This separates the measuring system from the DUT while the ESU is firing, and closes the connection between the DUT and ESU while the PSD-30’s fundamental frequency of 350 kHz alternating current is being applied. While the foot pedal is still being pressed, the microcontroller opens the ESU relays and closes the measuring relays. This now allows for safe and accurate measurements. When this switch happens, the microcontroller sends a signal to the DAQ to tell LabVIEW to measure the voltage drop across the reference resistor and the DUT. LabVIEW takes these voltage measurements and converts them into a resistance, using the known current of the system, and decides whether or not there has been too much thermal damage. From here, the microcontroller tells LabVIEW to stop measuring, and the ESU and measure relay switch orientation again. The ESU continues to send an alternating current to the DUT, and the process is repeated over and over again until the calculated resistance reaches some threshold value. Once this threshold is reached, LabVIEW sends a signal to the microcontroller to interrupt this process and turn off the ESU firing process. This ultimately stops the DUT from receiving any more alternating current from the ESU.

### 2.2. Validation of Feedback System

Before the validation can begin, a few programming considerations need to be completed. The first consideration that was made includes adding a delay of 25 ms between the transition of opening and closing the relays that separate the measuring system and ESU from the DUT. This is done to ensure that no current is sent back into the measuring system destroying the equipment. The 25 ms delay was chosen as a safe postponement time as the maximum switching time for the measure and ESU relays, according to the datasheet, is 20 ms.

This allows for a 5 ms buffer in case something goes wrong in the validation stage. An additional consideration had to be made in terms of a 25 ms delay between closing the measure relays and sending the signal to the DAQ to begin measuring. This fixed an error that occurred where the resistance measured would be extremely high as the initial measurements were made while the relays were still slightly open, during the switching of the relay. The last consideration that had to made concerns the ESU. The minimum amount of time that the ESU needed to be on before being turned off was found to be 75 ms. If the on-time was any less than this, then the ESU would display an error ‘Er P’ on the front panel. This error is caused by the ESU thinking that the P cord of the device is broken. With all of these considerations, the time to complete one full cycle of measuring the DUT, turning on and off the ESU, and back to measuring takes 0.184 s. This means that the ESU fires 5.4 times a second, with a total on-time of 0.405 s, giving the system a duty cycle of about 41%. A complete timing diagram is shown in Figure 6. The ‘Foot Pedal Pressed’ signal indicates wither the microcontroller reads a digital low, No, or digital high, Yes, signal from the modified foot pedal assembly. The ‘Measure Relays’ and ‘ESU Relays’ signals indicate when the relays are open or closed. The ‘ESU Firing’ signal indicates when the PSD-30 is applying an alternating current to the DUT. While the ‘LabVIEW Measure’ signal indicates when the impedance of the DUT is being measured.

The DUT used for validation is an RRC circuit which is the equivalent electrical circuit (Figure 7). used to mimic the human body [16]. Where the resistance of the ICW is denoted by R_I_, the resistance of the ECW is denoted by R_E_, and the cell membrane is denoted by C_M_. The values chosen are such that they represent the Cole-Cole curve from the ground patch location to the colon. An ImpediMed SFB7 (ImpediMed Ltd., Pinkenba, Australia) was used to determine these values. The SFB7 is capable of taking 256 measurements ranging from 4 kHz to 1 MHz in the time span of just about one second [20]. The device was connected at the same location that the ground patch would be on the thigh with the other end attached to the torso where the colon is located. The average of three measurement values for the front of the torso to the thigh was calculated to be 62.06 Ω for R_0_ and 25.35 Ω for R_∞_. While the average of three measurement values for the back of the torso to the thigh was calculated to be 65.50 Ω for R_0_ and 25.28 Ω for R_∞_. The R_0_ and R_∞_ values of these two positions were then averaged to give a close representation of what the Cole-Cole curve would look like if the snare were near the colon. The average values of these were calculated to be 63.78 Ω for R_0_ and 25.32 Ω for R_∞_. With these design specifications, the RRC circuit was created with a 64 Ω resistor for R_E_, a 18 Ω resistor for R_I_, and a 1nF capacitor for C_M_. As the measurements are all taken at the 50 kHz frequency and the equivalent circuit does not change values with the applied alternating current, a potentiometer was placed in series with the RRC circuit. This allows for the change in impedance of the DUT, by way of turning the potentiometer, to simulate the increase that would be observed if a tissue sample were being measured as an alternating current from the ESU was being passed through it.

Multiple tests of the feedback system were performed on the DUT with impedance threshold values set to 500 Ω, 750 Ω, and 1000 Ω. A low power setting on the ESU was used as this was the proof of concept step and the electrical components used were not rated for high wattage. In each of the trial runs, the alternating current from the ESU was stopped at each impedance threshold. During each trial, the impedance measurements from the reference resistor stayed within ±5 Ω of the known resistance of 475 Ω. The time between data points was also found to be the correct time delay of 0.184 s as predicted. The power to impedance curve for the 1000 Ω threshold trial is shown in Figure 8. As the coagulation switch was pressed on the modified foot pedal assembly, the potentiometer was turned simultaneously. The impedance started at 68 Ω and moved up to the 1000 Ω threshold where the ESU was automatically stopped by the feedback system, where the power dropped off to 0 W. Now that the feedback system has been verified as working as designed, the next stage of the study can be completed.

## 3. Results

After validating the impedance-controlled feedback system to operate as intended, the RRC equivalent circuit values were changed such that they represent the Cole-Cole curve of the body; R_I_ is 1.21 kΩ, R_E_ is 619 Ω, and C_M_ is 1 nF. This calculates to an R_0_ value of 619 Ω at zero frequency and an R_∞_ value of 409.5 Ω at infinite frequency. This is different than the values used in the validation stage where the operation was the main focus. These values were chosen to be the same values used in the RRC test cell included with the ImpediMed SFB7 to verify the accuracy of the results. With these values, a closer representation of the full picture of the whole body could be performed such that the dominating change in impedance did not come from the tissue sample alone. If the vales were not changed, the small resistor values used in the validation stage would be trumped by the larger impedance values seen by the tissue samples. This is not the case in a real-life scenario where the impedance of the full body of the patient is greater than the small area of the polyp. The potentiometer from the validation stage could also now be replaced with biological tissue samples. The tissue used was from a porcine colon, cut into 1 in × 1 in (2.54 cm × 2.54 cm) square samples. This sample was then placed in line with the feedback system by way of a 3M Red Dot electrode (3M, St. Paul, MN, USA) connected to the ground patch, and the snare pressed on top of the porcine colon sample (Figure 9).

The Red Dot electrode was chosen due to the ease of connecting the ground patch wire coming from the ESU to it, as well as replicating the gel grounding pad that is normally attached to the thigh of the patient during monopolar electrosurgery. Note that during a real-life situation, the grounding pad attached to the thigh is a much further distance than that of this model and impedance values would differ, however, the RRC circuit is used to counter this. The addition of the RRC circuit with the porcine colon sample is to simulate the full body so that the change in impedance from the tissue sample is a better representation and to handle this problem.

The feedback system now needs to be programmed to determine at what point perforation ensues and stop the alternating current from the ESU before this occurs. As the size of the patient and the polyp vary, a finite impedance threshold, used during the validation stage to stop the ESU needs to be abandoned in favor of a more dynamic approach. This new method will be based on the percent change in impedance. This approach is more appropriate as the initial conditions vary in each situation.

To collect both the prefire and postfire data that was needed to assess how much the impedance of the porcine colon sample has changed, due to the thermal damage, manually measuring with an oscilloscope was not feasible. During the initial trials, the time required to take multiple accurate measurements with an oscilloscope was found to take too long. As the sample sat on the gel pad in the open, the water in the sample started to evaporate and dry out the sample. This caused a noticeable change in impedance over a short period of time and skewed the data. This is where the use of the ImpediMed SFB7 comes into play once again. With its capability of taking 256 measurements of the reactance, resistance, phase, and impedance ranging from 4 kHz to 1 MHz in the time span of just about one second make this ideal. The SFB7 was used to take three measurements before and three measurements after the applied current. MATLAB R2017a (MathWorks, Natick, MA, USA) and Excel 2016 (Microsoft Corp, Redmond, DC, USA) were used to analyze the data.

### 3.1. Perforation Measurements

To determine what the critical percent change in impedance value should be, tests need to be performed to first determine the minimum time required for perforation to occur. This was done at the highest power setting of 50 W to determine the maximum range for the rest of the trials.

Visual inspection of the porcine colon sample, as well as the overall shape change of the traditional Cole-Cole curve, was performed to determine at what point perforation has occurred, as the ESU was fired multiple times for different durations. The initial values for this sample were 999 Ω for R_0_ and 523 Ω for R_∞_. The exact values for each of the firing durations for the zero and infinite frequency values are not needed during this preliminary experiment, as the test is to determine the maximum parameters for the coming trials. In Figure 10, the Cole-Cole curves were plotted, along with the tissue samples, for a firing duration of 2 s (a), 3 s (b), and 4 s (c) at the 50W setting. For the 2 s test, the Cole-Cole curve still supported the characteristics of not having much protein denaturation with both ICW and ECW as the measurements still show the traditional Cole-Cole curve as the points follow the fitted curve, with clear changes in reactance. Although after two seconds it is evident that at the low frequencies, the curve has started to level out. The fitted curve shows an R_0_ of about 1060 Ω with an R_∞_ value of just under 600 Ω. Visual inspection exposed no signs of perforation. For the 3 s test, the protein denaturation increased along with the hydro-change of the intracellular and extracellular water. This trend is expected, as the more thermal energy introduced burst more cell membranes deeper in the tissue. After three seconds, the fitted curve shows R_0_ close to 2000 Ω while R_∞_ approaches 0 Ω. Visual inspection of this sample also showed no signs of perforation. Moving on to the 4 s test, the Cole-Cole curve displayed clear signs of damage as the traditional curve has been completely lost. This shows that at all frequencies measured, the reactance did not change much. This means that extreme protein denaturation has occurred as many cell membranes have burst. There is no R_0_ or R_∞_ value to report as the fitted line did not cross the x-axis. Visual inspection showed that indeed perforation had occurred. Based on these graphs and visual inspections, perforation was deemed to occur around the 50 W power setting for a firing duration of 3 s. Therefore, this limit was chosen as the maximum parameters to test.

Visual inspection of the porcine colon sample, as well as the overall shape change of the traditional Cole-Cole curve, was performed to determine at what point perforation has occurred, as the ESU was fired multiple times for different durations. The initial values for this sample were 999 Ω for R_0_ and 523 Ω for R_∞_. The exact values for each of the firing durations for the zero and infinite frequency values are not needed during this preliminary experiment, as the test is to determine the maximum parameters for the coming trials. In Figure 10, the Cole-Cole curves were plotted, along with the tissue samples, for a firing duration of 2 s (a), 3 s (b), and 4 s (c) at the 50 W setting. For the 2 s test, the Cole-Cole curve still supported the characteristics of not having much protein denaturation with both ICW and ECW as the measurements still show the traditional Cole-Cole curve as the points follow the fitted curve, with clear changes in reactance. Although after two seconds it is evident that at the low frequencies, the curve has started to level out. The fitted curve shows an R_0_ of about 1060 Ω with an R_∞_ value of just under 600 Ω. Visual inspection exposed no signs of perforation. For the 3 s test, the protein denaturation increased along with the hydro-change of the intracellular and extracellular water. This trend is expected, as the more thermal energy introduced burst more cell membranes deeper in the tissue. After three seconds, the fitted curve shows R_0_ close to 2000 Ω while R_∞_ approaches 0 Ω. Visual inspection of this sample also showed no signs of perforation. Moving on to the 4 s test, the Cole-Cole curve displayed clear signs of damage as the traditional curve has been completely lost. This shows that at all frequencies measured, the reactance did not change much. This means that extreme protein denaturation has occurred as many cell membranes have burst. There is no R_0_ or R_∞_ value to report as the fitted line did not cross the x-axis. Visual inspection showed that indeed perforation had occurred. Based on these graphs and visual inspections, perforation was deemed to occur around the 50 W power setting for a firing duration of 3 s. Therefore, this limit was chosen as the maximum parameters to test.

### 3.2. Prefire vs Postfire Measurements

Now that the point of perforation has been discovered, further test using multiple power settings and firing durations was conducted to observe how the impedance changed. The power settings used ranged from 10 W–50 W over firing times of 1 s–3 s. This range was chosen based on the maximum power and duration found previously to be the cap. The measured quantities include the *DUT* resistance *R_DUT_*, reactance *X_c_*, and phase θ of the *DUT*. The resistance *R* on the x-axis is given by Equation (1) and the reactance on the y-axis is given by Equation (2):(1)$$R=R_{DUT}\times cos\left(\theta \right)$$
(2)$$\;X_{C}=R_{DUT}\times sin{\left(\theta \right)}_{\;}$$

Data points (256) for each trial are collected ranging from 4 kHz–1 MHz, although only 132 of these points are used in the calculations to determine the R_0_ and R_∞_ values. These 132 data points used to correlate to the measurements ranging from 25 kHz to 500 kHz. It is common to use this smaller range as it has been proven that measurements below and above this frequency range are normally not representative of the real picture [16]. Of these 132 data points, only 14 equally spaced points are plotted on the graph to allow for a clear visual understanding of what the graphs represent without being too cluttered. Because the amount of force the snare applied to the tissue sample also contributes to the thermal damage, a scale was used to measure the force applied and was adjusted to be the same for all tests ranging from 0.04–0.05 N. It should be noted that the change in impedance is based on two combined models, that of the RRC and porcine colon sample. This means that the minimum values should be that of the RRC model which describes the full-body given by an R_0_ value of 619 Ω and an R_∞_ value of 409.5 Ω. While the tissue sample should add to the overall impedance and correlate with the measured values above that of the RRC model.

One example of the multiple trials (30 W, 3 s) is shown in Figure 11 to demonstrate how the Cole-Cole curve changes from prefire to postfire. The left curve is the prefire measurement while the right curve is the postfire. The average of three measurements is plotted with the error bars representing the minimum and maximum values. In this example, the prefire Cole-Cole curve showed an R_0_ value of 988 Ω with an R_∞_ value of 717 Ω, while the 50 kHz measurement was 902 Ω. For the corresponding postfire Cole-Cole curve, R_0_ was 1221 Ω while R_∞_ was 855 Ω and the 50 kHz measurement was 1092 Ω. In this particular trial, the 50 kHz impedance measurement increased by 21%.

After performing the test for each power setting and firing duration, the 50 kHz frequency measurement for each trial was extracted from the data and used for the analysis. The details of each test are displayed in Figure 12. The average of each of the three measurements is plotted, with the error bars representing the maximum and minimum measurement. Investigating this graph shows a clear upward trend in the increase in impedance based on the duration of the firing as well as the power setting. For the 1 s tests, each of the power settings showed similar minimal increases in the percent of impedance, with the 50 W setting having the greatest increase with 7.36%. For the 2 s tests, the gap between each power setting starts to increase. The lowest percent increase in impedance was that of the 10 W trial with 6.68%, while the largest increase came from the 50 W setting with an increase of 19.92%. For the 3 s tests, the gap increased further with the 10 W trial having an increase in impedance of 10.09% and the 50 W trial increasing by 27.39%.

As mentioned before, the 50 W, 3 s firing duration was deemed to be the limit as perforation ensues. This appears to happen at an increase in impedance of 27.4% while the second-highest increase of 24.4% occurs at a power setting of 40 W with a firing duration of 3 s. With these two trials being the greatest increase in impedance, and knowing the 50 W 3 s trial caused perforation while the 40 W 3 s trial did not, a 25% increase in impedance was chosen as the interrupt point. This 25% increase was then programmed into the LabVIEW virtual instrument (VI).

### 3.3. Impedance Over Time

After implementing the 25% increase in impedance threshold into the VI, multiple tests of the impedance-controlled feedback system were performed for different power settings on the ESU ranging from 10 W–50 W. During the initial trials, it was found that the firing time of 75 ms, found to be the minimum allowed time for the ESU to work properly without showing errors, was too short to apply an effective alternating current to the porcine colon tissue sample. With this short firing time, no thermal damage was being detected, resulting in no change of impedance. To overcome this matter, the firing time was increased to 500 ms. With this longer application of alternating current, the thermal damage ensued was detected and measured over time. The results for each test are displayed in Figure 13, Figure 14, Figure 15, Figure 16 and Figure 17. The data points represent the impedance as calculated by LabVIEW and the dotted lines between the data points represent the period of time that the ESU was firing. There are no data points during these times, as no measuring of the DUT takes place during the firing cycle. While measurements of the DUT were taking place, the impedance measured across the reference resistor was monitored and observed to stay within ±5 Ω of the ideal 475 Ω. The solid horizontal line represents the cut-off threshold of the ESU, which is a point defined as a 25% increase from the initial impedance measurement.

For the 10 W power trial, the initial impedance was measured to be 1612 Ω which corresponds to a cut-off threshold of 2015 Ω. The total time, from the beginning of the first firing cycle of the ESU to the first measurement taken above the threshold, took 15.092 s. The 20 W power trial had a starting impedance of 1281 Ω with a calculated threshold point of 1601 Ω. The elapsed time to reach the threshold point took 11.534 s. With the ESU set to 30 W, the trial started with an impedance of 1271 Ω resulting in a threshold value of 1589 Ω. The time to reach this threshold took 5.994 s. The next trial, with a power setting of 40 W, took 6.137 s to make it from the initial impedance of 1491 Ω to the threshold point of 1864 Ω. With the highest power setting of 50 W, the initial impedance was measured to be 1152 Ω resulting in a threshold of 1440 Ω. While the elapsed time to reach this threshold took 4.123 s.

## 4. Discussion

After observing the impedance vs time graphs for each of the trials, the alternating current from the ESU was stopped after an increase in impedance of 25% or more was measured. This shows the successful implementation and execution of the device as-built and programmed. Further observation of the impedance vs time graphs demonstration that the impedance measurements drop sharply after the first firing cycle and increase over time. This is thought to occur from the bursting of the cell membranes around where the snare touches the porcine colon sample. During the first firing cycles of each trial, it was visually observed that a small amount of water would pool around the snare. This would lower the impedance as water is a good conductor and the ratio of water to tissue is high. As the firing cycles continued, the water was visually observed to evaporate. As more of the water from the busted cell membranes evaporate, the measured impedance values also increased as the tissue denaturalization increased. This trend continued until the pool of surface water was minimized from the thermal energy being applied. Once this happened, the measured impedance increased beyond the starting impedance and damage to the tissue followed. This thought is further backed by observing the length of time the measured impedance stayed below the initial measurement for each power setting. As the power is increased, the applied current density increases and in turn an increase in the rate of applied thermal energy. This was not observed during the prefire vs postfire test as the applied current was steady and was not applied via pulses.

First, examine the 10 W trial. The measured impedance for this trial stayed below the initial impedance for almost 14 s and took 22 firing cycles before reaching the threshold point. For the 20 W trial, the time the measured impedance stayed below the initial was only 7 s while requiring 16 firing cycles to reach the threshold point. Given a higher applied current density, the increase in thermal energy causes the water from the busted cell membranes to evaporate faster as well as tissue damage. This trend is further observed for higher power trials. For the 30 W and 40 W trials, the impedance stayed below the initial measurement for almost 5 s with each going through nine firing cycles. Note that although each of these trials took the same number of firing cycles to reach the threshold point, the 40 W trial ended slightly higher above the cut-off point. For the 50 W trial, the impedance only stayed below the initial measurement for less than 3 s while only requiring six firing cycles to reach the threshold point. Looking closely at the impedance vs time graphs for each trial shows that in some cases, the impedance threshold point was surpassed. This is evident in the 20 W, 40 W, and 50 W trials. As the power is increased, the amount of change in impedance for each 500 ms firing cycle is increased. For the 20 W trial, the last measurement was just below the cut-off point. This resulted in an additional firing cycle that increased the impedance beyond this point. For the 40 W and 50 W trials, the amount of impedance change per firing cycle was larger than the other trials. This was enough to surpass the threshold level as the measurements before this point was closer to the cut-off point than the increase per firing cycle.

While many electrosurgical device manufacturers have some version of an auto-stop system built-in, the downside to these devices, however, is the fact that the auto-stop systems are usually directed towards the bipolar method and with the use of forceps. Although there have been strides in designing a similar type of auto-stop system that implements impedance measurements using the monopolar method, this study used the hot biopsy forceps [21]. The advantage of the impedance-controlled feedback polypectomy device in this study is the ability to auto-stop using the monopolar method with a snare. This is important due to the fact that most polyps are classified as either small or diminutive, with snare polypectomy being the preferred method to remove polyps of these sizes. While the experience of the surgeon is a main contributing factor for complications such as perforation to occur, this auto-stop system should help to minimize excess thermal injury incurred by the patient. This device should allow the surgeon to perform their job more safely and with higher confidence and in turn, lead to fewer complications during surgery with higher success rates.

The limitations of this device include the inability to measure the impedance of the sample simultaneously as the ESU is applying current. If this hurdle is crossed, then the device would have the ability to stop closer to the threshold point without going over. Such was the case for the 20 W, 40 W, and 50 W trials. This overshoot in the applied alternating current could in turn cause perforation to occur if it becomes too severe. Further limitations include the single frequency measurement. If the dependence on the ImpediMed device for sweeps in the frequency is overcome, then the ability to capture the full picture of the DUT including the R_0_ and R_∞_ values during firing could be obtained. This would allow for more precise programming to be developed instead of only the 50 kHz dependence post-capture that is currently in place.

In the future, we hope to gather in vivo data about the impedance values of various polyps during different times including before, during, and right before the removal of the polyp. This will allow for a better understanding of how the impedance changes of the polyp in a more realistic setting. With this data, the impedance-controlled device can be further programmed to overcome any unforeseen situations as the performance of the device will change in a real use case. It is important to note that although the device is expected to perform differently in vivo, the programming has been designed to be easily modified. One way in the future to help overcome the problem of high overshoots when operating the feedback system could be the implementation and use of overshoot tolerance bands. The idea of these tolerance bands would be to monitor when the increase in impedance is approaching the threshold point, before surpassing it and resulting in a perforation. They could work in a variety of ways. One way could be by turning off the firing operation prematurely if the impedance was measured to be within a certain percentage close enough to the threshold point. This would prevent overshoot from occurring by not allowing the impedance to actually reach the threshold point. Another way to implement tolerance bands could be to lower the firing duration of the ESU as the impedance approached closer to the threshold point. The amount the impedance increases each firing cycle is dependent on the duration of applied current from the ESU. This would lower the overshoot by reducing the amount the impedance increases between each measuring cycle. These bands could also be adjusted for each power setting. The 10W trial increased less each time the ESU fired than the 50 W trial. This would mean that the overshoot tolerance bands could be closer together for the lower power settings as well as be farther apart for the higher power settings. This would ensure that at any power setting, the impedance could increase closer to the threshold value while minimizing as much overshoot as possible. Also in the future, we plan to avoid using the ImpediMed device totally and implement the use of the LabVIEW VI to control the function generator to sweep through the frequencies necessary while measuring to capture the Cole-Cole curve.

## 5. Conclusions

The design and implementation of this impedance-controlled hot snare polypectomy device were shown to accurately measure and monitor the change in impedance, with an accuracy of ±5 Ω, of a porcine colon sample ex vivo. This allowed for the system to interrupt and stop the high-frequency alternating current from the ESU being applied to the sample before perforation occurred. This was accomplished by determining when the impedance of the sample had increased from its’ initial measurement by 25%. The intent of this impedance-controlled device is such that it be used as an add-on device or accessory to complement the current device a surgeon may have. If the foot pedal assembly is similar to that of the PSD-30’s, then this device has the capability to work. This will keep the costs down for the surgeon, as there is no need to purchase another ESU, and ultimately the patient. It also allows for the use of the current and preferred snares and ground patches, so no additional or new accessories need to be purchased.

## Figures and Tables

**Figure 1 sensors-20-00142-f001:**
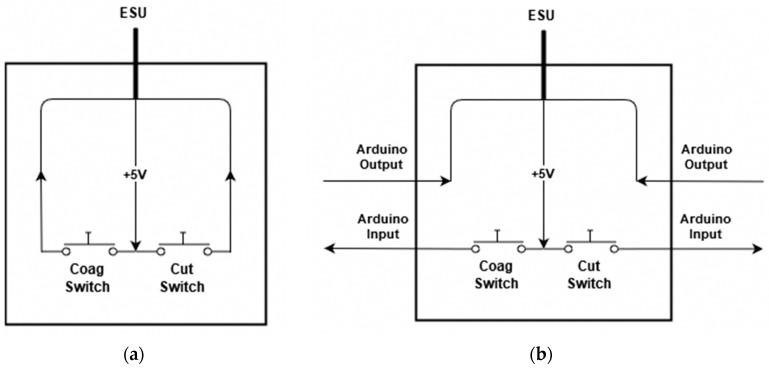
(**a**) Unmodified foot pedal assembly and (**b**) modified foot pedal assembly.

**Figure 2 sensors-20-00142-f002:**
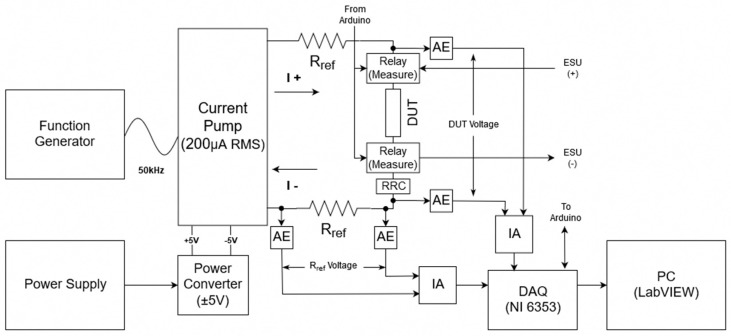
Measuring system component of feedback system.

**Figure 3 sensors-20-00142-f003:**
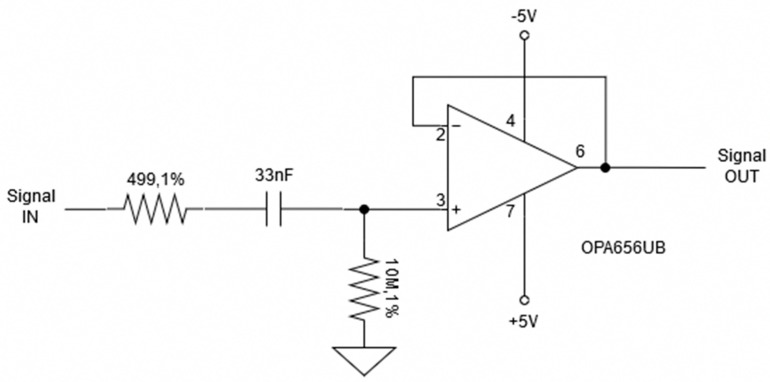
Active electrode used to stabilize signal for measurements.

**Figure 4 sensors-20-00142-f004:**
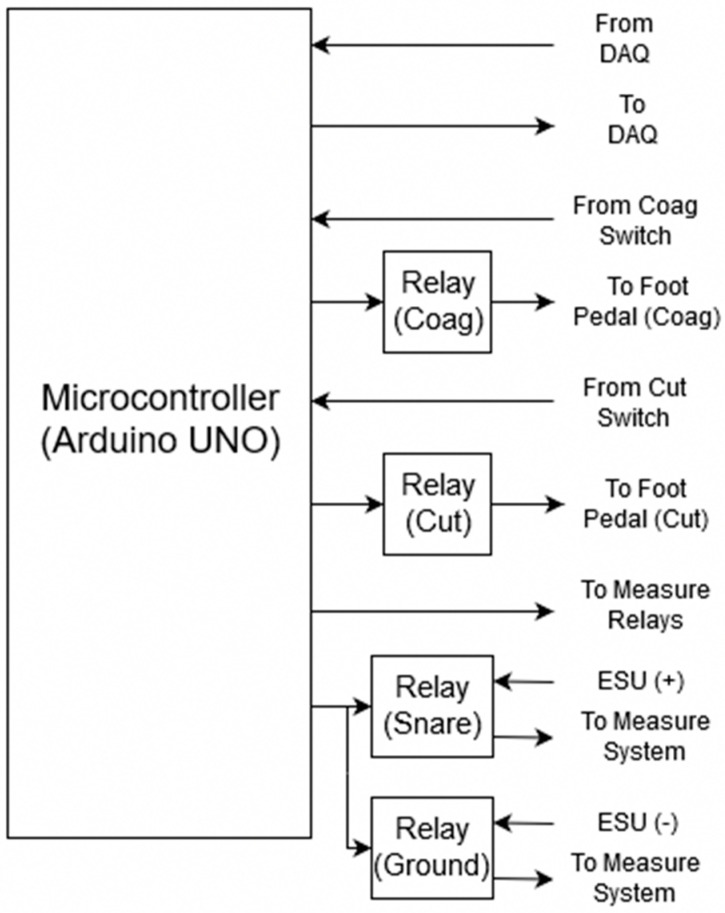
Control system component of feedback system.

**Figure 5 sensors-20-00142-f005:**
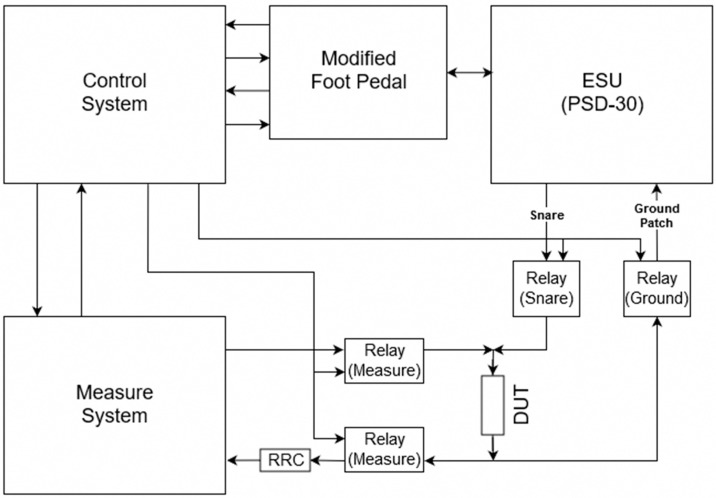
Diagram of the complete impedance-controlled feedback system.

**Figure 6 sensors-20-00142-f006:**
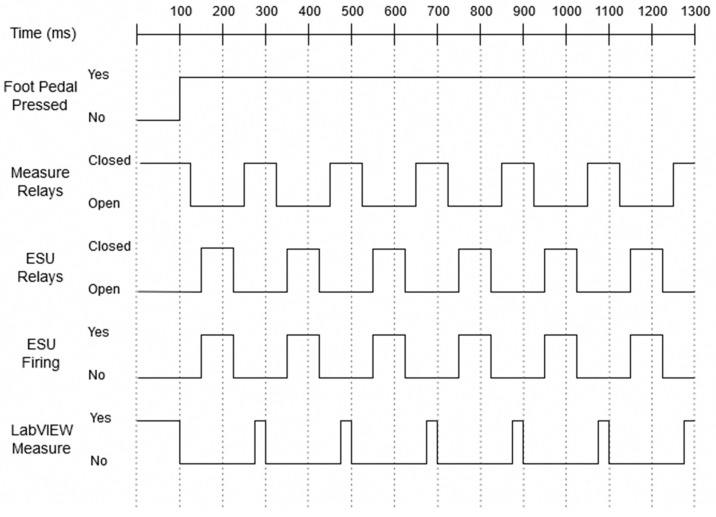
Timing diagram of feedback system operations.

**Figure 7 sensors-20-00142-f007:**
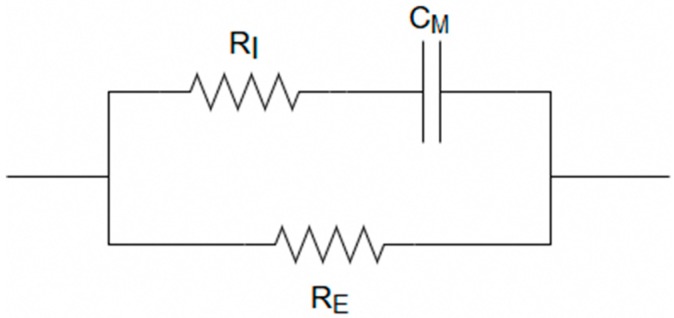
Rrc equivalent circuit used in series with the dut.

**Figure 8 sensors-20-00142-f008:**
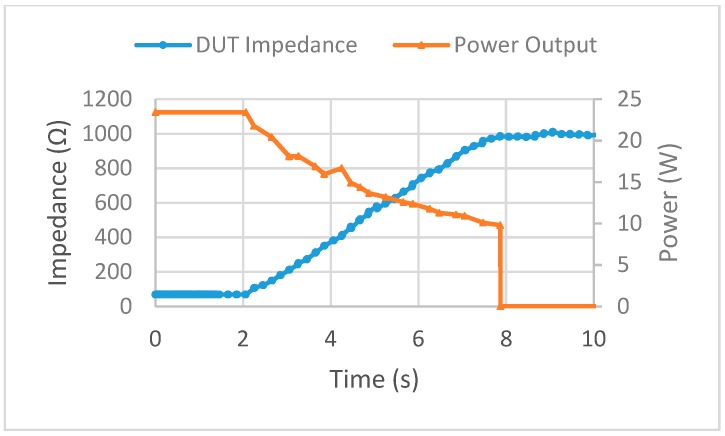
Power to impedance curve for 1000 Ω threshold validation stage.

**Figure 9 sensors-20-00142-f009:**
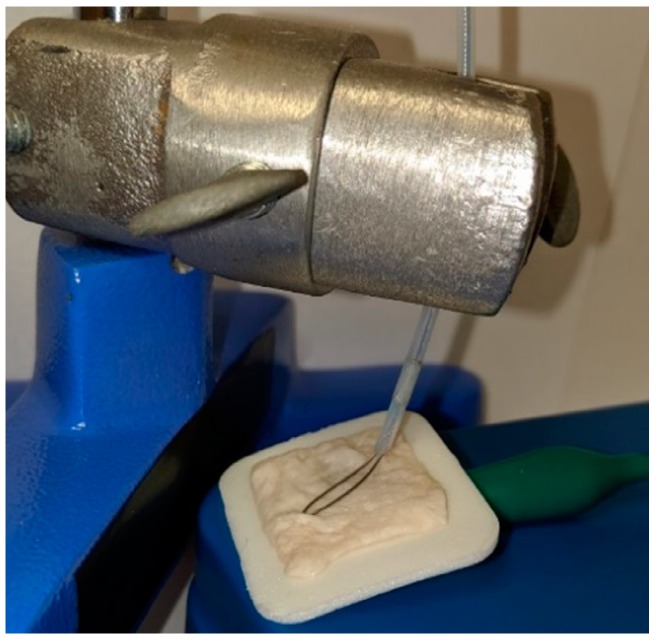
Tissue sample setup of snare and porcine colon.

**Figure 10 sensors-20-00142-f010:**
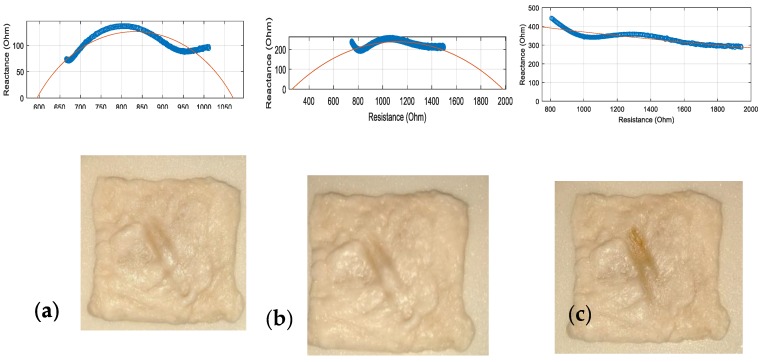
Perforation test at 50 W power setting for (**a**) 2 s (**b**) 3 s and (**c**) 4 s. The circles represent the data points and the solid line is the fitted curve.

**Figure 11 sensors-20-00142-f011:**
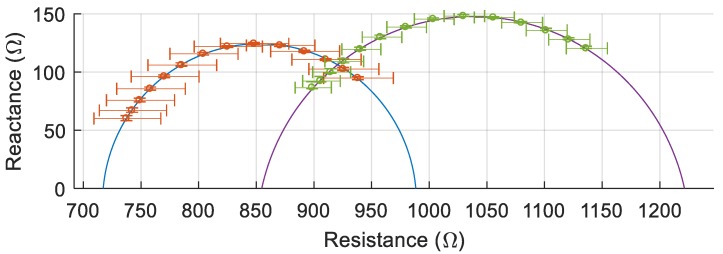
Pre vs postfire Cole-Cole plot at 30 W for 3 s. The circles represent the data points and the solid line is the fitted curve.

**Figure 12 sensors-20-00142-f012:**
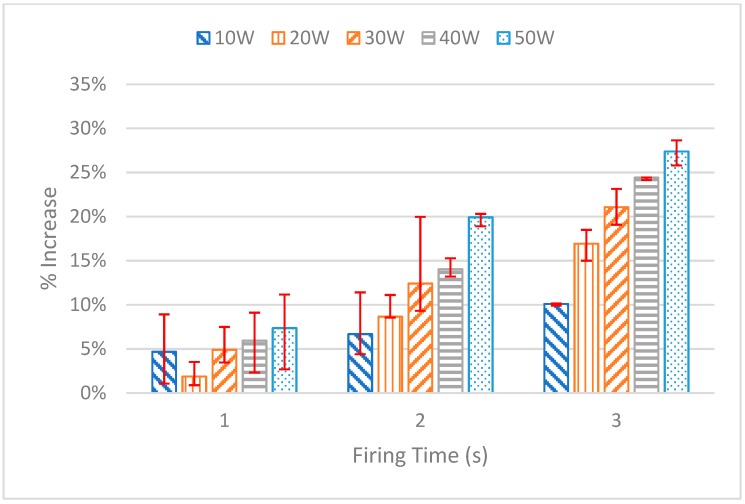
Pre vs postfire increase in impedance for the 50 kHz data points.

**Figure 13 sensors-20-00142-f013:**
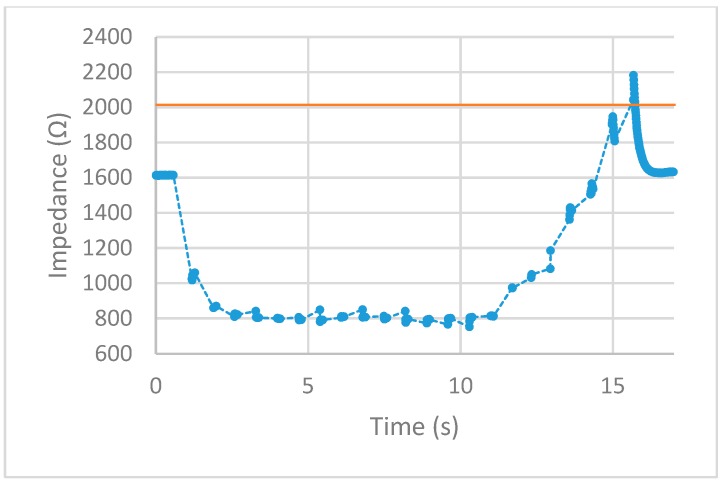
10 W test. circles represent the measured impedance, dashed lines represent the firing time, and solid line represents the cut-off threshold.

**Figure 14 sensors-20-00142-f014:**
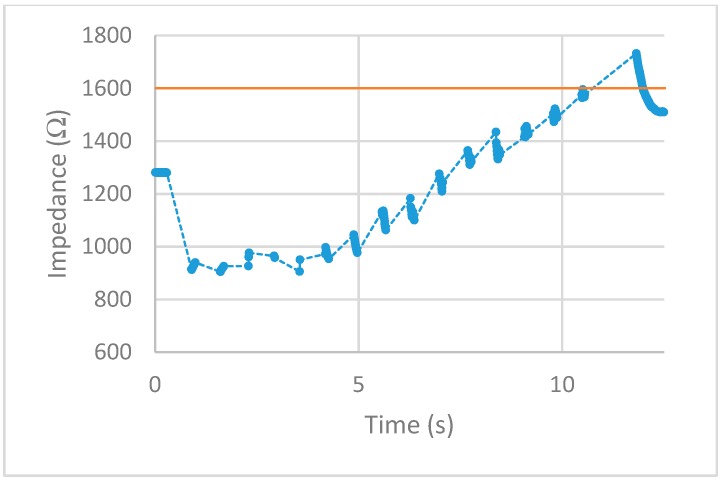
20 W test. circles represent the measured impedance, dashed lines represent the firing time, and solid line represents the cut-off threshold.

**Figure 15 sensors-20-00142-f015:**
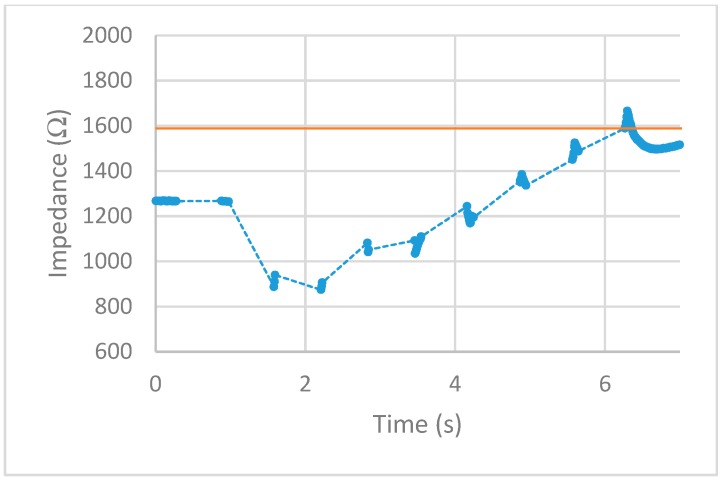
30 W test. circles represent the measured impedance, dashed lines represent the firing time, and solid line represents the cut-off threshold.

**Figure 16 sensors-20-00142-f016:**
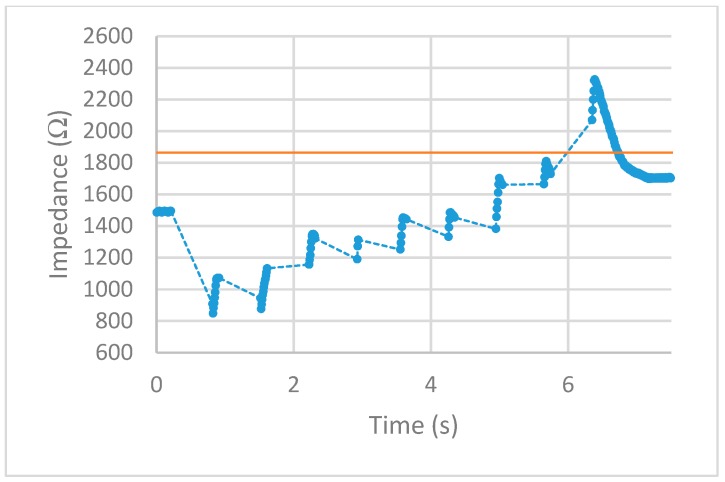
40 W test. circles represent the measured impedance, dashed lines represent the firing time, and solid line represents the cut-off threshold.

**Figure 17 sensors-20-00142-f017:**
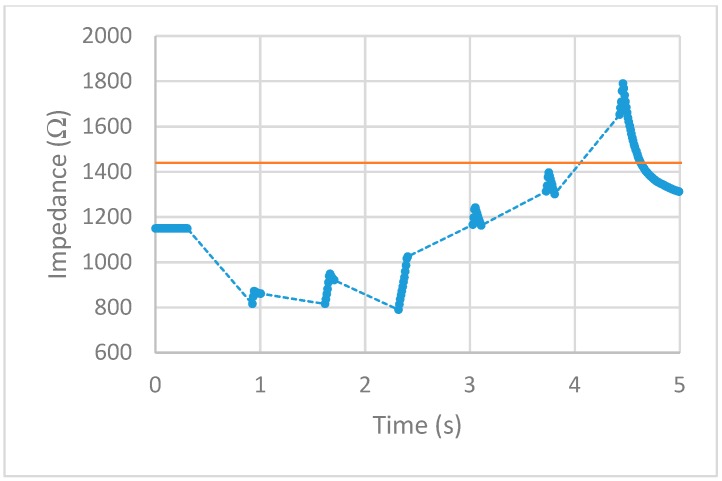
50 W test. circles represent the measured impedance, dashed lines represent the firing time, and solid line represents the cut-off threshold.

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
