# Peer review of "Design of an Impedance-Controlled Hot Snare Polypectomy Device"

_sensors, 2019, doi:10.3390/s20010142_

Round 1
Reviewer 1 Report
Article of interest, but has important weaknesses in the method and approach, so it must be rejected. Even so, it is recommended to redo the work and its re-submission.
1) The detailed features of the commercial ESU must be provided in a section or table.
2) A figure with a descriptive diagram of the modifications made to the ESU must be incorporated. A detailed description of the modifications supported by the figure must be made. The need for each element and its functionality must be explained. As example: the need and functionality of the PYB10-Q24-D5 DC-DC converter has not been adequately explained, the need and functionality of the Howland current pump, operational amplifiers, relays, etc. has not been explained.
3) A figure with a descriptive diagram of the measuring system must be incorporated. A detailed description of the elements of the measuring system must be made based on the new figure. The need and functionality of different elements that make up the measuring system must be provided.
4) A figure with a descriptive diagram of the controlling system must be incorporated. A detailed description of the elements of the controlling system must be made based on the new figure. The need and functionality of different elements that make up the controlling system must be provided.
5) A new complete system image must be included with all its elements: modified ESU, measuring system and controlling system. In the text and referring to that image a better explanation of the interconnection of the elements is convenient.
6) An image of the timing used in the measurement process is also convenient. The different intervals must be identified by parameters, and a more detailed explanation of these intervals, their functionality and characteristics must be made in the text based on the figure.
7) On line 125, page 4, it is mentioned that the B&K Precision model 4053 Function Generator is used. It should be noted that the problem of current control in the measurement has not been addressed, since when setting the voltage, the injected current will depend on the impedance of the biological medium. Later it is commented that a current of 200uA is used. The detailed circuit diagram must be provided, explaining the way in which the current is set. The way in which the effect of the 50 ohm output impedance of the function generator is avoided must also be considered. As a detailed explanation of the circuit diagram has not been made, these issues have not been adequately resolved.
8) On line 125, page 4, the selection of the reference resistance of 475 ohms must be justified.
9) On line 124, page 4, Figure 2 is not suitable for explaining the voltage measurement. This measure must be explained including a complete image. The use of two Rref must be justified.
10) On line 130, page 4, the following is mentioned: “frequencies from about 5kHz to 500kHz need to be measured”. This statement must be justified.
11) On line 179, page 5, the following is mentioned: “350kHz alternating current is being applied.” The origin of this frequency must be justified.
12) On line 212, page 6 the following is mentioned: “body; RI is 1.21kΩ, RE is 619Ω, and CM is 1nF. “The parameters of this model must be justified.
13) In Figure 5 the experimental setup is shown, which differs greatly from a real situation. A biological tissue sample of 2.54cm x 2.54cm is employed, but the thickness of the sample is not indicated. This model collects only the effect of surface impedance, and does not take into account the current path from the snare to the external electrode. Such impedance can vary greatly, so that a thorough analysis of this limitation must be performed. The position of the external electrode in a real situation should be provided.
14) In Figure 6 the bioimpedance values must be provided before applying current. A description of the bioimpedance values before applying the current must be incorporated in the text.
15) The model in Figure 3 describes one dispersion. The zero and infinite frequency resistance values have been obtained from a model with a single dispersion (probably the ImpediMed device model). If Figure 6 is observed, in the bioimpedance measures the main dispersion can be seen, but also other dispersions at low and high frequency, which may be due to parasitic effects or circumstances typical of the measurement procedure. The identification of resistance parameters at zero and infinite frequency should be carried out in the main dispersion, which would lead to very different parameters from those mentioned in the work. The description of the evolution of the zero and infinite resistance parameters is wrong.
16) The damage assessment (perforation) must be adequately justified. The method considered suggests a visual inspection, but a more adequate justification should be performed.
17) The procedure to estimate the impedance value must be described in detail. As it seems that phase information is not used, it seems that the measurement technique calculates the impedance module. Equations should be shown, properly identifying the parameters used. Figure 7 represents only resistance variations. It should be clarified if the impedance module is actually calculating the resistance value. In case it is the module, the 25% increase with respect to the resistance value must be justified.
18) In Figures 9-13, the reason why the impedance is stable during the first moments of time must be explained, as well as the existence of discontinuities between measured values (dashed lines).
19) Line 45, page 2. “The most commonly encountered, over 80% - 90%, polyps sizes found during…” Repeated: encountered, found
Reviewer 2 Report
This paper describes a modification to a ESU to measure impedance during the use of a hot snare polypectomy device. Whilst there are similar methods one some other devices, the authors have identified the need for better control using a hot snare to prevent perforation. They present a 2 channel impedance sensor which measures the impedance between a reference electrode and the hot snare between firing of the ESU. The work is promising, particularly that is works as a potential add on to existing devices, however there are some points which require addressing before it is ready for publication.
The written description of the system is in too much detail, but is lacking an overview diagram/schematic which would make it easier to understand the connections.
How does this device change the efficacy of the hot snare? The authors state a duty cycle of 41% and minimum run times, but not if this changes the how the hot snare would work in practice. In particular, are these changes appropriate from a clinical point of view? Would this require the changing of habits of the surgeon?
The threshold of 25% was based on the phantom experiments, where the 2nd reference electrode is very close to the hot snare in the colon sample, where in reality it would be placed on the outside surface of the patient. This would increase the overall impedance measured by the device, and thus lower the percentage change seen. This is because the localised impedance change of the colon wall makes a smaller contribution to the apparent impedance. Therefore, there needs to be a method of dynamically changing the threshold, or better understanding of the localised impedance change. Electrical Impedance Tomography methods have been suggested for ablation for instance.
The authors suggest that a single frequency is necessary as sweeps are too slow, however the spectra contains important information that is lost at a single frequency. Especially as the cole-cole plots are used to determine the threshold duration. If the cole-cole parameters were obtained each measurement, then the damage could be assessed directly, instead of using an inferred threshold value. Could the ImpediMed device be set to collect less frequency points so as to collect impedance spectra during the hot snare use? As the authors are using an arbitrary function generator, it would be relatively simple to generate a multi frequency input signal and collect data at multiple frequencies simultaneously.
The actual testing of the device is limited to a few samples, which are far from the real use case, some consideration of how the performance of the device would change in reality is necessary in the discussion.
What is happening to cause the rise in impedance between firing cycles in the figures9-12? Is this a result of tissue properties changing or heating? Or is it an effect of the electrode charging due to charge imbalance?
The manuscript is laid out coherently, and the writing explains the results well. However the style drifts into colloquialism at some points and in sometimes has a story telling tone of a report.
The figure captions are not very informative and overall the figures could be improved as they are lacking polish.
Round 2
Reviewer 1 Report
Q1: The detailed features of the commercial ESU must be provided in a section or table.
A1: Added to “Development of Feedback System” section; line 98.
Q1 (AFTER THE REVIEW): The description must be more detailed and extensive, specifying the full operation of the modes of operation.
Q2: A figure with a descriptive diagram of the modifications made to the ESU must be incorporated. A detailed description of the modifications supported by the figure must be made. The need for each element and its functionality must be explained. As an example: the need and functionality of the PYB10-Q24-D5 DC-DC converter have not been adequately explained, the need and functionality of the Howland current pump, operational amplifiers, relays, etc. has not been explained.
A2: Added to “Development of Feedback System” section; starting line 107; added figures 1 and 2. Current pump line 132; op-amp line 137; active electrodes line 164; instrumentation amplifies line 165; relays starting line 187.
Q2 (AFTER THE REVIEW): The schematic and circuit diagram of the active electrode (AE).
Q3: A figure with a descriptive diagram of the measuring system must be incorporated. A detailed description of the elements of the measuring system must be made based on the new figure. The need and functionality of different elements that make up the measuring system must be provided.
A3: Added to “Development of Feedback System” section; starting line 125; added figure 3
Q3 (AFTER THE REVIEW): Ok
Q4: A figure with a descriptive diagram of the controlling system must be incorporated. A detailed description of the elements of the controlling system must be made based on the new figure. The need and functionality of different elements that make up the controlling system must be provided.
A4: Added to “Development of Feedback System” section; starting line 178; added figure 4
Q4 (AFTER THE REVIEW): Ok
Q5: A new complete system image must be included with all its elements: modified ESU, measuring system and controlling system. In the text and referring to that image, a better explanation of the interconnection of the elements is convenient.
A5: Added figure 5
Q5 (AFTER THE REVIEW): Ok
Q6: An image of the timing used in the measurement process is also convenient. The different intervals must be identified by parameters, and a more detailed explanation of these intervals, their functionality and characteristics must be made in the text based on the figure.
A6: Added to “Validation of Feedback System” section; starting line 240; added figure 6
Q6 (AFTER THE REVIEW): Ok
Q7: Online 125, page 4, it is mentioned that the B&K Precision model 4053 Function Generator is used. It should be noted that the problem of current control in the measurement has not been addressed since
when setting the voltage, the injected current will depend on the impedance of the biological medium. Later it is commented that a current of 200uA is used. The detailed circuit diagram must be provided, explaining the way in which the current is set. The way in which the effect of the 50-ohm output impedance of the function generator is avoided must also be considered. As a detailed explanation of the circuit diagram has not been made, these issues have not been adequately resolved.
A7: The function generator sends the signal into the current pump which keeps the injected current constant; line 134. Effect of 50-ohm output impedance has been addressed; line 145. Due to a patent problem, the detailed circuit diagram of the current pump itself cannot be provided.
Q7 (AFTER THE REVIEW): Ok
Q8: Online 125, page 4, the selection of the reference resistance of 475 ohms must be justified.
A8: Added; line 160
Q8 (AFTER THE REVIEW): Ok
Q9: Online 124, page 4, Figure 2 is not suitable for explaining the voltage measurement. This measure must be explained including a complete image. The use of two Rref must be justified.
A9: Added figure 3.
Q9 (AFTER THE REVIEW): Ok
Q10: Online 130, page 4, the following is mentioned: “frequencies from about 5kHz to 500kHz need to be measured”. This statement must be justified.
A10: Added; line 152
Q10 (AFTER THE REVIEW): Ok
Q11: Online 179, page 5, the following is mentioned: “350kHz alternating current is being applied.” The origin of this frequency must be justified.
A11: Added; line 212
Q11 (AFTER THE REVIEW): It has not been answered in that line.
Q12: On line 212, page 6 the following is mentioned: “body; RI is 1.21kΩ, RE is 619Ω, and CM is 1nF. “The parameters of this model must be justified.
A12: Added; line 290
Q12 (AFTER THE REVIEW): The introduction and justification of the body model is not clearly explained.
Q13: In Figure 5 the experimental setup is shown, which differs greatly from a real situation. A biological tissue sample of 2.54cm x 2.54cm is employed, but the thickness of the sample is not indicated. This model collects only the effect of surface impedance, and does not take into account the current path from the snare to the external electrode. Such impedance can vary greatly so that a thorough analysis of this limitation must be performed. The position of the external electrode in a real situation should be provided.
A13: The RRC circuit in series with the porcine colon sample is used to represent the full body. The impedance of the tissue sample is not the only value in play. This has been added and discussed starting line 299.
Q13 (AFTER THE REVIEW): The introduction and justification of the body model is not clearly explained.
Q14: In Figure 6 the bioimpedance values must be provided before applying current. A description of the bioimpedance values before applying the current must be incorporated in the text.
A14: The bioimpedance values are not needed during this preliminary test as the main focus is to determine the maximum parameters to use for the following test. Discussed starting line 331.
Q14 (AFTER THE REVIEW): The reviewer does not agree with the response, since the initial values provide reference information on the bioimpedance values before applying the electric current.
Q15: The model in Figure 3 describes one dispersion. The zero and infinite frequency resistance values have been obtained from a model with a single dispersion (probably the ImpediMed device model). If Figure 6 is observed, in the bioimpedance measures the main dispersion can be seen, but also other dispersions at low and high frequency, which may be due to parasitic effects or circumstances typical of the measurement procedure. The identification of resistance parameters at zero and infinite frequency should be carried out in the main dispersion, which would lead to very different parameters from those mentioned in the work. The description of the evolution of the zero and infinite resistance parameters is wrong.
A15: Added; See starting line 368
Q15 (AFTER THE REVIEW): The reviewer does not agree with the response and continues to make the same comment, referred in this case to Figure 10. Line 368 and the following lines do not provide an adequate response.
Q16: The damage assessment (perforation) must be adequately justified. The method considered suggests a visual inspection, but a more adequate justification should be performed.
A16: The overall shape change in the traditional cole cole plot was also used. See discussion starting line 329.
Q16 (AFTER THE REVIEW): Line 329 and the following lines do not provide an adequate response.
Q17: The procedure to estimate the impedance value must be described in detail. As it seems that phase information is not used, it seems that the measurement technique calculates the impedance module. Equations should be shown, properly identifying the parameters used. Figure 7 represents only resistance variations. It should be clarified if the impedance module is actually calculating the resistance value. In case it is the module, the 25% increase with respect to the resistance value must be justified.
A17: Added; See starting line 355
Q17 (AFTER THE REVIEW): Line 355 and the following lines do not provide an adequate response
Q18: In Figures 9-13, the reason why the impedance is stable during the first moments of time must be explained, as well as the existence of discontinuities between measured values (dashed lines).
A18: discontinuities between measured values (dashed lines), see starting line 415; the reason why the impedance is stable during the first moments of time, see starting line 453.
Q18 (AFTER THE REVIEW): Line 415, line 453 and the following lines do not provide an adequate response.
Q19: Line 45, page 2. “The most commonly encountered, over 80% - 90%, polyps sizes found during…” Repeated: encountered, found
A19: Removed ‘found’; line 47
Q19 (AFTER THE REVIEW): Ok
Q20: Figures 10, 11, 12, 13, 14, 15, 16, 17 must clarify in the caption or inside the figure the meaning of the different traces.
Reviewer 2 Report
The authors have addressed the points about the clarity of the manuscript, but the experiments are insufficient to fully test the device. I would encourage the authors to continue working with this device with more realistic experiments as I think the concept is a sound one. If the cole cole plots are what is necessary to set the threshold, then having a direct measure would be more robust to noise and inter/intra patient variation.
Q1: these figures are helpful but there is still a lot of excessive detail in this section. Fig 1 and 2 are essentailly the same.
Q2: I appreciate the contact force was controlled, but I was asking whether there was evidence that the device had altered the performance, or whether the clinicians agrred the threshold was suitable based n their experience
Q3: I had missed the point of the RRC, however this is a fixed single impedance. In reality this would drift and change during surgery and be different between patients. So a percentage threshold is too simplistic a measure. More consideration and experiments showing the threshold is valid for different values off RRC and different samples are needed.
Q4: I appreciate the pains of using “black box” devices that can’t be changed! However, with the function generator and the sampling rate of the ni daq, you already have a system capable of performing frequency sweeps and thus collecting the full cole cole plots, even with reduced snr to make the sweeps fast enough.
Q5: the does not address the insufficient samples adequately
Q6: quoted line does not match the text. These measurements are taken between firings, is the water evaporating when the firings are not taking place?
Q8: the captions are still very uninformative, so the figures cannot be understood without reading the text at the same time.
Round 3
Reviewer 1 Report
All comments have been adequately addressed. The work is suitable for publication.
Author Response
Thank you so much for your review efforts.
Reviewer 2 Report
The authors have addressed the points raised, I would suggest another pass at making the methodology clearer and adding more detailed discussion of the limitations and how they are going to be addressed in the future.
